# Unexpected Tension Pneumothorax after Double-Lumen Endotracheal Intubation in Patients with Pulmonary Edema: A Case Report

**DOI:** 10.3390/medicina59030460

**Published:** 2023-02-25

**Authors:** Jongyoon Baek, Sang Jin Park, Myungjin Seo, Eun Kyung Choi

**Affiliations:** Department of Anesthesiology and Pain Medicine, Yeungnam University College of Medicine, 170, Hyeonchung-ro, Nam-gu, Daegu 42415, Republic of Korea

**Keywords:** tension pneumothorax, double-lumen endotracheal intubation, complication, airway exchange catheter, case report

## Abstract

Tension pneumothorax is a relatively rare complication after anesthetic induction that requires prompt diagnosis and treatment. Several handling errors related to intubation procedures or equipment and vigorous positive pressure ventilation are potentially important etiologies of tension pneumothorax in patients with underlying lung disease or in mechanically ventilated patients. We describe a case of tension pneumothorax observed after double-lumen tube (DLT) insertion followed by single-lumen tube replacement using an airway exchanger catheter in a mechanically ventilated patient. An 84-year-old female on mechanical ventilation underwent minimally invasive cardiac surgery under general anesthesia. Immediately after left-sided DLT insertion using an airway exchanger catheter, oxygen saturation decreased to 89%, peak airway pressure increased to 35 cm H_2_O with inadequate tidal volume, and blood pressure gradually dropped to 69/41 mmHg. Breath sounds from the right hemithorax were significantly reduced. Severe collapse of the right lung, a flattened diaphragm, and compressed abdominal organs were identified on chest radiography. Therefore, a tube thoracotomy was performed based on the findings of a tension pneumothorax. Then, oxygen saturation, peak airway pressure with adequate tidal volume, and blood pressure improved, and the distended abdomen normalized. After the pneumothorax resolved, a bronchoscopy was performed. Slight redness was noted in the right bronchus, indicating that the DLT was incorrectly inserted into the right side. In conclusion, the possibility of a tension pneumothorax should be considered during DLT intubation or endotracheal tube replacement with an airway exchange catheter.

## 1. Introduction

Tension pneumothorax is a relatively rare complication after anesthetic induction that requires prompt diagnosis and treatment. Numerous factors lead to tension pneumothorax following endotracheal intubation, including multiple intubation attempts; inadequate size, depth, and cuff pressure of the endotracheal tube; tube malposition; and inadequate use of a stylet or airway exchange catheter. In particular, several handling errors related to intubation procedures or equipment and vigorous positive pressure ventilation are potentially important etiologies of tension pneumothorax in patients with underlying lung disease or in mechanically ventilated patients [1].

We present a case of unexpected tension pneumothorax, which is suspected to have resulted from incorrect insertion of a left-sided double-lumen tube (DLT) into the right bronchus while a single-lumen tube was replaced by a DLT using an airway exchanger catheter in a mechanically ventilated patient with pulmonary edema who underwent minimally invasive cardiac surgery.

## 2. Case Presentation

An 84-year-old woman (height: 158 cm; weight: 58 kg) was referred to our institution for mitral valve prolapse. The patient had a history of hypertension and diabetes mellitus. Chest radiography revealed bilateral pulmonary edema and effusion. Transthoracic echocardiography revealed a left ventricular ejection fraction (LVEF) of 76% and mitral valve prolapse. During hospitalization, she was treated with torsemide, nitrates, and beta-blockers; however, her dyspnea and pulmonary edema worsened.

Oxygen saturation (SpO_2_), checked with O_2_ at 5 L/min via a reservoir mask, was 89%. She was intubated with an endotracheal tube (size 7.0), and mechanical ventilation was initiated. Mechanical ventilation was set at an inhalation oxygen fraction of 0.4, a tidal volume (TV) of 400 mL, a respiratory rate of 14/min, a minute volume of 5.6 L/min, and a positive end-tidal pressure (PEEP) of 7 cm H_2_O. Frothy, pinkish secretions in the endotracheal tube required frequent suction. A transesophageal echocardiogram showed severe eccentric mitral regurgitation (flail mitral valve leaflet, mainly A2) due to chordal rupture, LVEF (80%) with mild aortic regurgitation, and mild pulmonary hypertension (right ventricular systolic pressure, 46 mmHg). Mitral valve replacement was scheduled, and a minimally invasive approach was planned under left one-lung ventilation with DLT. The surgery was scheduled for two days after mechanical ventilation was initiated to allow time to administer furosemide to improve pulmonary edema. Repeat chest radiography the day before surgery showed cardiomegaly with consolidation in both lung fields, however, the consolidation was slightly improved (Figure 1).

On the day of surgery, the patient was taken to the operating room in an intubated state with a single-lumen tube. When the patient was admitted to the operating room, she was under moderate sedation with dexmedetomidine and remifentanil infusion and had an arterial line in the left radial artery and a central venous catheter in the left internal jugular vein. Vital signs before anesthesia induction were as follows: blood pressure (BP), 101/61 mmHg; heart rate, 90 bpm; and SpO_2_, 100%. Electrocardiography revealed a normal sinus rhythm. General anesthesia was induced with 1 vol% sevoflurane, 0.2 µg/kg/min remifentanil infusion, and 50 mg rocuronium. After the loss of spontaneous breathing, mechanical ventilation was initiated with an inhalation oxygen fraction of 1.0, a TV of 400 mL, and a respiratory rate of 12/min. The end-tidal CO_2_ was 30–32 mmHg, and the peak airway pressure (Paw) was 19 cm H_2_O. The single-lumen tube was changed to a 35 Fr left-sided DLT (Mallinckrodt™ double-lumen tube, Mallinckrodt Medical, Athlone, Ireland) using an 11 Fr airway exchange catheter (Cook^®^ airway exchange catheter, Cook Medical, Bloomington, IN, USA). Before disconnecting the circuit from the tube, an initial assessment was performed using a GlideScope^®^ videolaryngoscope (Verathon Medical, Bothell, WA, USA), and a grade 1 (Cormack and Lehane) laryngoscopic view was seen. The airway exchange catheter was gently advanced until the endotracheal tube depth was approximately 2 cm above the carina. Maintaining a view of the glottis with a GlideScope^®^, the 35 Fr DLT was advanced through the catheter to a depth of 26 cm. Auscultation revealed that both lung sounds were audible. However, SpO_2_ suddenly decreased from 99% to 89%, and Paw increased to 35 cm H_2_O with an inadequate TV. The BP decreased from 105/43 mmHg to 69/41 mmHg, so a phenylephrine bolus was administered, and norepinephrine and epinephrine infusions were started. The patient’s abdomen was severely distended (Figure 2a). We removed the DLT and reintubated a size 7.0 single-lumen tube using a direct laryngoscope. BP, SpO_2_, Paw, and TV were maintained with mechanical ventilation (pressure-control mode) at 105/52 mmHg, 89%, 32 cm H_2_O, and 270 mL, respectively. We placed a nasogastric tube and attempted decompression of the abdomen; however, this did not change the clinical situation. Upon auscultation, the breath sounds on the right were significantly reduced. We immediately checked the portable chest radiograph on suspicion of a right-sided tension pneumothorax. Chest radiography revealed a tension pneumothorax with severe collapse of the right lung, a flattened diaphragm, and compression of the abdominal organs (Figure 3a). After tube thoracotomy, Paw, TV, SpO_2_, and BP were 25 cm H_2_O, 400 mL, 98%, and 143/66 mmHg, respectively, and the epinephrine infusion was discontinued. The distended abdomen resolved (Figure 2b). Repeat portable chest radiography revealed that the pneumothorax had resolved (Figure 3b). On bronchoscopy using a flexible fiberscope, there was slight redness in the right bronchus, suggesting that the DLT was incorrectly inserted into the right side. However, there was no visible mechanical injury. A right minithoracotomy was performed in a semilateral position, and mitral valve replacement was performed under cardiopulmonary bypass. After mitral valve replacement, we attempted to wean from the bypass. However, despite our efforts to wean the patient from bypass, hemodynamic instability persisted due to poor heart function. Finally, extracorporeal membrane oxygenation (ECMO) was performed, and the patient was transferred to the intensive care unit, still intubated. On postoperative day 1, the patient died because vital functions could not be maintained despite ECMO support.

## 3. Discussion

Tension pneumothorax immediately after anesthetic induction is a relatively rare complication with potentially catastrophic outcomes. Once tension pneumothorax occurs, sudden hypoxemia followed by hypotension is observed, and it often has a malignant course, leading to circulatory arrest. Therefore, prompt recognition and treatment of tension pneumothorax are essential to minimize morbidity and mortality.

Surgical or anesthesia-related factors, including central venous cannulation and regional block in proximity to the pleura, positive pressure ventilation, and incorrect endotracheal intubation, can induce a pneumothorax [1,2,3,4]. Tension pneumothorax related to airway manipulation, such as endotracheal intubation, has been shown to result in poorer outcomes [5]. Therefore, some recommendations for safe endotracheal intubation include selecting an adequate tube size, avoiding deep introduction and advancement against resistance, slow inflation of the tracheal cuff with adequate pressure, judicious use of the stylet, and manual ventilation with adequate positive pressure [6]. Concerning DLT insertion, choosing the correct size and depth of the tube is important to avoid airway trauma and prevent barotrauma [7]. A large DLT can rupture the respiratory tract, while small tubes can result in barotrauma due to deep insertion. The left main bronchus diameter can be used as a guide to determine the appropriate tube size [8]. The correct tube insertion depth can be confirmed by fiberoptic bronchoscopic examination. In this case, a 35 Fr DLT was used with a tube depth of 26 cm. A 35 Fr DLT is generally used in women in our institution, and a depth of 26 cm is not deep when based on the patient’s height [9]. However, when DLT was inserted, we could not use fiberoptic visual guidance because the single-lumen tube was changed via an airway exchange catheter, and cardiovascular collapse with hypoxemia occurred immediately before confirming the tube position with a fiberoptic bronchoscope.

In this case, vigorous positive pressure ventilation in the right bronchial intubated state may have led to the tension pneumothorax. High inspiratory peak pressure after double-lumen intubation and fiberoptic bronchoscopic imaging (some erosion spots near the carina and right main bronchus) support this scenario. Most pneumothoraxes in intubated patients are associated with underlying lung diseases, such as pneumonia, chronic obstructive pulmonary disease, and acute respiratory distress syndrome and a tension pneumothorax is more common in ventilated patients [1]. Previous studies have shown that a high incidence of barotrauma is associated with the underlying process of low compliance [10]. In patients with underlying lung disease, dependent lung lesions have low compliance and require high airway pressures. The maldistribution of mechanical TV promotes gas trapping and uneven alveolar distention, with a risk of pneumothorax. In addition, subtle subpleural bullae or intrapulmonary air cysts, which can occur in acute lung disease, may cause a pneumothorax. In the present case, the patient underwent mechanical ventilation for 3 days preoperatively because of pulmonary edema and a pleural effusion. Therefore, her preexisting lung condition might have made her susceptible to high inspiratory peak pressures immediately after DLT insertion, which might have predisposed her to a tension pneumothorax.

Another concern, in this case, was related to the use of an airway exchange catheter for difficult reintubation (from a single-lumen tube to a DLT). Endotracheal tube exchange is challenging for anesthesiologists and requires vigilance. Although an airway exchange catheter is an effective and secure way to improve tube exchange safety, potential severe complications in the two categories of barotrauma and airway perforation remain a concern. Barotrauma, derived from jet ventilation or oxygen supplementation via the hollow core of an airway exchange catheter, accounts for up to 11% of the cases with 50 psi jet ventilation [11]. Oxygen delivery via an airway exchange catheter should be assessed in terms of airway patency and expiration guarantee [12]. Safeguards such as mid-tracheal placement and a pressure limiting valve can reduce adverse events related to high-pressure supplement oxygen. Perforation of the tracheobronchial tree is rare and is associated with higher morbidity, pneumothorax, pneumomediastinum, and even death [12]. Although the airway exchange catheter was carefully inserted to the target depth estimated from preoperative radiographs and previous studies [13], the possibility of tracheobronchial perforation due to unknown anatomical reasons should be considered in the clinical setting. The manufacturer’s guidelines for the Cook^®^ airway exchange catheter recommend the application of a sterile lubricant, marking the targeted depth on the catheter, and maintaining the tip 2–3 cm above the carina. In adults, an airway exchange catheter should be introduced no further than 26 cm without resistance [14]. In this case, after single-lumen tube reintubation with adequate mechanical ventilation, we could not find any laceration or perforation of the tracheobronchial wall; however, some erosion spots near the carina and right main bronchus were observed when visualized with the fiberoptic bronchoscope. Although we could not rule out more distal bronchial injuries (further fiberoptic bronchoscopy was not performed), a culprit lesion was never found.

We can use a DLT or bronchial blocker to isolate the lung or selectively ventilate. Depending on the clinical situation, each lung isolation method has its own set of advantages. Bronchial blockers are best used in children with airways that are too small for a DLT. They allow selective lobar isolation, have less potential for airway trauma, and can be placed through an endotracheal tube or laryngeal mask airway. In contrast, a DLT can be placed more quickly and easily than a bronchial blocker; intraoperative tube displacement occurs less frequently; it allows suctioning before reinflation; and continuous positive airway pressure is more easily applied [15]. In this case, we also considered using bronchial blockers, but a DLT was used because the patient was in a high secretion state due to pulmonary edema and because of the difficulty of lung deflation when using bronchial blockers. Moreover, because of the short distance from the right upper bronchus to the carina, correct placement of bronchial blockers may be more difficult, whereas bronchial blockers may be more easily dislodged. Therefore, isolation of the right lung for minimally invasive mitral valve surgery may be limited.

With regards to the surgical approach, minimally invasive mitral valve surgery has demonstrated its advantages: reduced bleeding and transfusion, decreased risk of atrial fibrillation and sternal wound infection, shorter ventilator support time, and shorter hospitalization [16]. However, despite the superior perioperative outcomes of this technique compared to conventional sternotomy, its application is limited due to the exposure difficulties of the mitral valve and the risk of cerebrovascular accidents [16]. Moreover, when the patient has a respiratory problem, minimally invasive mitral valve surgery requiring one-lung ventilation may increase the risk of intraoperative hypoxemia [17]. In this case, the patient showed a slight improvement in pulmonary edema with preoperative diuretic administration, and the surgeon selected this surgical method in consideration of its advantages over its risks. In addition, transcatheter mitral valve repair with MitraClip™ (Abbott Laboratories, Abbott Park, IL, USA) might be a possible therapeutic option; this intervention has emerged in selected patients with mitral regurgitation at high surgical risk for excessive comorbidities, advanced age, or decreased left ventricle function [18]. However, transcatheter intervention with MitraClip™ could not be applied in our institution due to the limitations of its introduction and use.

## 4. Conclusions

In conclusion, physicians should consider the possibility of tension pneumothorax during DLT intubation or endotracheal tube change with an airway exchange catheter. Patients with underlying lung disease or those who are mechanically ventilated, in particular, require a more cautious approach than other patients because the risks associated with these procedures can have disastrous consequences.

## Figures and Tables

**Figure 1 medicina-59-00460-f001:**
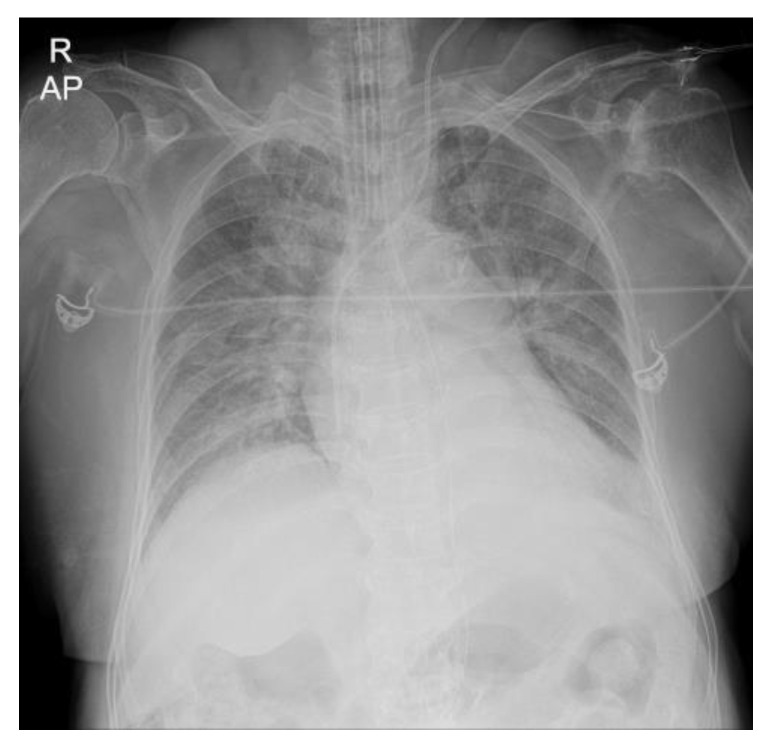
Preoperative chest radiography. Both lung field consolidations, indicating pulmonary edema, were demonstrated.

**Figure 2 medicina-59-00460-f002:**
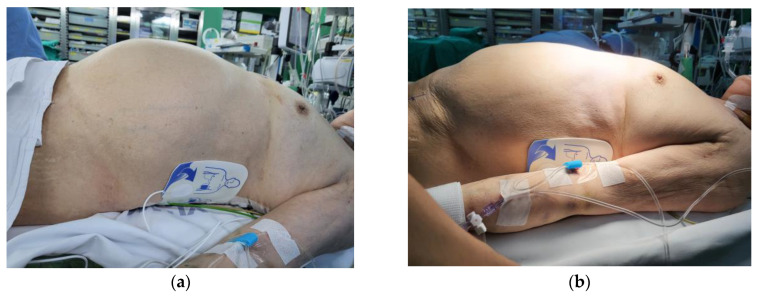
(**a**) The patient’s abdomen became distended. (**b**) The patient’s abdomen became soft and flat after the tube thoracotomy.

**Figure 3 medicina-59-00460-f003:**
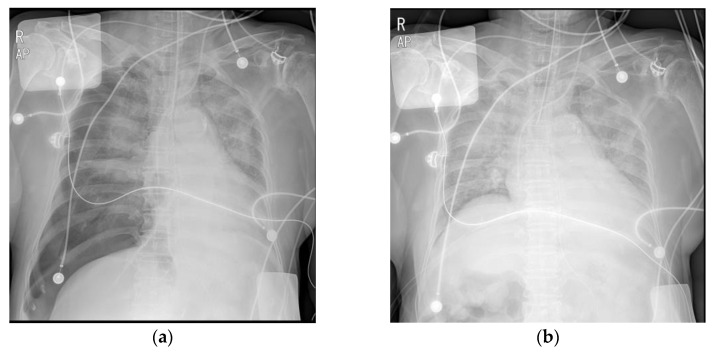
(**a**) The right lung field showed tension pneumothorax in the operating room. (**b**) After the tube thoracotomy, the pneumothorax was improved.

## Data Availability

Not applicable.

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
