# Peer review of "Unexpected Tension Pneumothorax after Double-Lumen Endotracheal Intubation in Patients with Pulmonary Edema: A Case Report"

_medicina, 2023, doi:10.3390/medicina59030460_

Round 1

Reviewer 1 Report

Dear Authors, 

Thank you for the opportunity to review the case report entitled "Unexpected tension pneumothorax after double-lumen endotracheal intubation in patients with pulmonary edema: A case report". The article points out the risk of a severe complication after DLT intubation, which is pneumothorax. The manuscript is written in the clear and correct language. Nevertheless, I do not find it to be of sufficient merit for publication in the Medicina. 

Major issues:

1.It remains unclear why a patient with pulmonary edema (or ARDS? in the publication, these terms appear interchangeably, see line 135) to miniMVR. Didn't you expect problems with oxygenation during OLV? 

2.Did the Heart Team consider placing a mitraClip in the patient? (If not available, transfer of the patient to a reference center may be considered)

3.Why was it decided to insert a DLT without fiberoptic guidance when one was available?

4.Why was ultrasound not used to diagnose pneumothorax? (Ultrasound appears to be standard equipment in the operating theater) 

5.How was the postoperative period?

Author Response

Jan 24th, 2023

Dear reviewer,

We appreciate the detailed, supportive and constructive review given to our manuscript, entitled “Unexpected Tension Pneumothorax after double-lumen endotracheal intubation in patients with pulmonary edema: A case report” (medicina-2138456). Here, we submit a first revised version with a point-to-point reply to your comments. All changes made in the text were highlighted in red. Again, thank you for your consideration and we look forward to hearing from you.

Eun Kyung Choi MD. PhD

Response to Reviewer Comments

Point 1: It remains unclear why a patient with pulmonary edema (or ARDS? in the publication, these terms appear interchangeably, see line 135) to miniMVR. Didn't you expect problems with oxygenation during OLV?

Response 1: Compared to conventional open surgery, miniMVR has advantages such as reduced bleeding and transfusion, decreased risk of atrial fibrillation and sternal wound infection, reduced ventilator support time and postoperative pulmonary complications, shorter intensive care unit and hospital stay, and increased patient satisfaction [1–3]. Therefore, miniMVR is generally practiced for mitral valve replacement in our institution. This patient had pulmonary edema, but it was slightly improved with administration of a diuretic before surgery.

We changed line 154 from ARDS to ‘pulmonary edema and a pleural effusion.’

Point 2: Did the Heart Team consider placing a mitraClip in the patient? (If not available, transfer of the patient to a reference center may be considered)

Response 2: MitraClip was not available in our hospital at that time because it was only available in very few hospitals in other regions in South Korea. So, it was impossible to transfer the patient and wait the intervention.

Point 3. Why was it decided to insert a DLT without fiberoptic guidance when one was available?

Response 3: Airway exchange catheters can be used to increase the safety of changing endotracheal tubes [4]. Therefore, we used an airway exchange catheter to insert a DLT.

After intubation, there was a sudden decrease in SpO2 and an increase in peak airway pressure. Concurrently, blood pressure dropped. Therefore, vasopressor administration and reintubation with a single lumen tube were performed without delay, and there was no chance for using a bronchoscope.

We added the description of this part in the discussion, page 4, line 137-140.

Point 4. Why was ultrasound not used to diagnose pneumothorax? (Ultrasound appears to be standard equipment in the operating theater)

Response 4: At that time, only the TEE probe could be used for the ultrasound in the operating room, and the ultrasound that could use the linear probe or convex probe was being used in other operating rooms. There was a portable x-ray that could be used in the operating room, so we checked chest radiograph right after taking it.

Point 5. How was the postoperative period?

Response 5: We described it in the case presentation part, page 3, line 106-111.

References

  1. Cao, C.; Gupta, S.; Chandrakumar, D. et al. A Meta-Analysis of Minimally Invasive versus Conventional Mitral Valve Repair for Patients with Degenerative Mitral Disease. Ann Cardiothorac Surg 2013, 2, 693–703
  2. Cheng, D.C.H.; Martin, J.; Lal, A. et al. Minimally Invasive versus Conventional Open Mitral Valve Surgery: A Meta-Analysis and Systematic Review. Innovations (Phila) 2011, 6, 84–103,
  3. Santana, O.; Reyna, J.; Grana, R. et al. Outcomes of Minimally Invasive Valve Surgery versus Standard Sternotomy in Obese Patients Undergoing Isolated Valve Surgery. Ann Thorac Surg 2011, 91, 406–410
  4. Benumof, J.L. Airway Exchange Catheters : Simple Concept, Potentially Great Danger. Anesthesiology 1999, 91, 342–344

Reviewer 2 Report

Dear authors,

the manuscript is written sufficiently and the length is acceptable. Please give your statement to the following points:

TEXT

-          We need to update the scientific literature to improve the discussion

-          please specify the aim of the study and clinical message that the authors want to send

References

-          Please check the journal’s guidelines

Best regards

Author Response

Jan 24th, 2023

Dear reviewer,

We appreciate the detailed, supportive and constructive review given to our manuscript, entitled “Unexpected Tension Pneumothorax after double-lumen endotracheal intubation in patients with pulmonary edema: A case report” (medicina-2138456). Here, we submit a first revised version with a point-to-point reply to your comments. All changes made in the text were highlighted in red. Again, thank you for your consideration and we look forward to hearing from you.

Eun Kyung Choi MD. PhD

Response to Reviewer Comments

Point 1: We need to update the scientific literature to improve the discussion.

Response 1: Considering your comments, we update the scientific literature and revised in the discussion part.

Point 2:  please specify the aim of the study and clinical message that the authors want to send.

Response 2: As your comments, we revised the introduction and conclusion part.

Point 3: References, Please check the journal’s guidelines.

Response 3: Considering your comments, we revised.

Round 2

Reviewer 1 Report

Dear Authots,

Thank you again for the opportunity to review the article. I still think that the article needs to be clarified for the reader (especially cardiothoracic anesthesiologists). First, I fully agree that minithoracotomy may have an advantage in a SELECTED group of patients, certainly not in a patient with underlying severe respiratory failure, where attempted SLV may lead to hypoxemia significantly increasing the risk of intraoperative and postoperative complications. I suggest raising the issue of mini procedure eligibility in the discussion (10.21037/jtd-20-1804). In addition, showing other than classical methods of MV repair (mitraClip) in the discussion seems reasonable from the point of view of the article's merit. 

Author Response

Feb 1st, 2023

Dear reviewer,

We appreciate the detailed, supportive, and constructive review given to our manuscript, entitled “Unexpected Tension Pneumothorax after double-lumen endotracheal intubation in patients with pulmonary edema: A case report” (medicina-2138456). Here, we submit a second revised version with a point-to-point reply to your comments. All changes made in the text were highlighted in red. Again, thank you for your consideration and we look forward to hearing from you.

Eun Kyung Choi MD. PhD

Response to Reviewer Comments

Reviewer 1:

Point 1: First, I fully agree that minithoracotomy may have an advantage in a SELECTED group of patients, certainly not in a patient with underlying severe respiratory failure, where attempted SLV may lead to hypoxemia significantly increasing the risk of intraoperative and postoperative complications. I suggest raising the issue of mini procedure eligibility in the discussion (10.21037/jtd-20-1804).

Response 1: As your comments, we added the issue related to minithoracotomy in last section of discussion, page 5, line 195-205 and added relating references (15-17).

Point 2: Second, showing other than classical methods of MV repair (mitraClip) in the discussion seems reasonable from the point of view of the article's merit. 

Response 2: As your comments, we added the issue of MitraClip in the last section of discussion, page 5, line 205-210.